# Evaluating the Translational Potential of Bacteriocins as an Alternative Treatment for *Staphylococcus aureus* Infections in Animals and Humans

**DOI:** 10.3390/antibiotics12081256

**Published:** 2023-07-30

**Authors:** Lauren R. Heinzinger, Aaron R. Pugh, Julie A. Wagner, Michael Otto

**Affiliations:** Pathogen Molecular Genetics Section, Laboratory of Bacteriology, National Institute of Allergy and Infectious Diseases, National Institutes of Health, Bethesda, MD 20814, USA; lauren.heinzinger@nih.gov (L.R.H.); aaron.pugh@nih.gov (A.R.P.); jalsmp1@gmail.com (J.A.W.)

**Keywords:** antibiotic alternatives, bacteriocins, *Staphylococcus aureus*, MRSA, animal studies, clinical trials

## Abstract

Antibiotic resistance remains a global threat to human and animal health. *Staphylococcus aureus* is an opportunistic pathogen that causes minor to life-threatening infections. The widespread use of antibiotics in the clinical, veterinary, and agricultural setting combined with the increasing prevalence of antibiotic-resistant *S. aureus* strains makes it abundantly clear that alternatives to antibiotics are urgently needed. Bacteriocins represent one potential alternative therapeutic. They are antimicrobial peptides that are produced by bacteria that are generally nontoxic and have a relatively narrow target spectrum, and they leave many commensals and most mammalian cells unperturbed. Multiple studies involving bacteriocins (e.g., nisin, epidermicin, mersacidin, and lysostaphin) have demonstrated their efficacy at eliminating or treating a wide variety of *S. aureus* infections in animal models. This review provides a comprehensive and updated evaluation of animal studies involving bacteriocins and highlights their translational potential. The strengths and limitations associated with bacteriocin treatments compared with traditional antibiotic therapies are evaluated, and the challenges that are involved with implementing novel therapeutics are discussed.

## 1. Introduction

Antibiotic resistance remains a global threat to human health. The World Health Organization predicts that, by 2050, the increase of antibiotic resistance could lead to an estimated 10 million deaths annually, costing approximately USD 100 trillion [1]. *Staphylococcus aureus*, specifically methicillin-resistant *S. aureus* (MRSA), is a significant contributor to this trend despite *S. aureus* being an opportunistic pathogen that only causes infection under certain circumstances. Many *S. aureus* infections occur in individuals who are already colonized by the bacterium. This is significant because 30–50% of the population is colonized with *S. aureus* on their skin, in their intestine, or in their nasal cavities; about one third of the population are persistently colonized [2,3]. Certain conditions also significantly increase an individual’s risk of developing a symptomatic *S. aureus* infection, such as type 1 diabetes [4], immunodeficiencies [5], and undergoing surgery [6] or hemodialysis [7]. Infections vary in presentation based on the site of infection but can include bacteremia, sepsis, toxic shock syndrome, endocarditis, pneumonia, gastroenteritis, meningitis, urinary tract infections, scalded skin syndrome, osteomyelitis, and multiple skin and soft tissue infections (SSTIs) [8]. To this day, *S. aureus* remains one of the most difficult-to-treat ESKAPE pathogens (*Enterococcus faecium*, *S. aureus*, *Klebsiella pneumoniae*, *Acinetobacter baumannii*, *Pseudomonas aeruginosa*, and *Enterobacter* species).

As patients infected with *S. aureus* tend to have recurrent infections, antibiotic resistance can quickly occur. Several major classes of antibiotics (e.g., β-lactams, glycopeptides, fluoroquinolones) are now largely ineffective at controlling *S. aureus* infections. A timeline describing the emergence of antibiotic-resistant strains is depicted in Figure 1. The methods by which *S. aureus* gains resistance and the impact that these resistance mechanisms have on treatment outcomes have been the focus of extensive research. Historically, the treatment of *S. aureus* infections primarily involved penicillin G, which is a β-lactam antibiotic. The production of β-lactamase led to the emergence of penicillin-resistant *S. aureus* (PRSA), and PRSA became widespread by the late 1950s. Synthesis of penicillin G derivatives that were resistant to β-lactamase hydrolysis became a priority, and methicillin, a β-lactam antibiotic that is resistant to hydrolysis by β-lactamase, was synthesized not long after. Methicillin functions similarly to other β-lactams as it targets the penicillin-binding proteins (PBPs) that are used in the formation of peptidoglycan in the cell wall [9]. Unfortunately, methicillin-resistant strains of *S. aureus* (MRSA) began to appear not long after methicillin was first used to treat *S. aureus* infections in the clinic [10]. Methicillin resistance occurs through the expression of PBP2a, a PBP that is resistant to methicillin’s mechanism of action and that can take over the cross-linking reactions of the host PBPs [11]. MRSA strains differ genetically from methicillin-sensitive *S. aureus* (MSSA) isolates by the presence of a large stretch of foreign DNA that includes the *mecA* gene, which encodes PBP2a. A global rise in MRSA infections and high resistance to all β-lactam antibiotics resulted in multidrug-resistant MRSA clones becoming the most common causative agent of *S. aureus* infections by the late 1990s [12]. Community-associated MRSA (CA-MRSA) began to spread in the 1990s [13], and livestock-associated MRSA (LA-MRSA) began to spread in the 2000s [14].

Glycopeptides such as vancomycin or teicoplanin are a class of antibiotics that eliminate *S. aureus* by preventing cell wall synthesis. Vancomycin specifically interferes with late-stage peptidoglycan synthesis, which results in cell death [15]. Glycopeptides and specifically vancomycin have remained the most common treatment for MRSA infections since the 1980s [16]. The use of vancomycin increased as a direct result of the increasing frequency of MRSA infections [17], which led to the emergence of vancomycin-resistant strains. Vancomycin-resistant *S. aureus* is categorized into vancomycin intermediate-resistance *S. aureus* (VISA) and vancomycin-resistant *S. aureus* (VRSA). The major difference between VISA and VRSA is the threshold in the minimum inhibitory concentration (MIC) values (4–8 µg/mL vs. ≥16 µg/mL, respectively). The first VISA strain was isolated from a patient in Japan in 1997 [18], but some studies have suggested that reduced susceptibility to vancomycin in *S. aureus* isolates dates back to 1987 [19]. The first VRSA strain was isolated from a patient in 2002 in the United States [20]. VISA phenotypes are known to arise from heterogenous populations of *S. aureus* where most cells have little or no resistance to vancomycin. After exposure to vancomycin, most *S. aureus* cells are eliminated, but some survive; these survivors persist and ultimately result in the homogenous VISA phenotype [21]. VRSA strains gain vancomycin resistance via the *vanA* operon on transposon Tn*1546*, which was originally obtained from a vancomycin-resistant enterococci (VRE) conjugative plasmid [22]. VRSA strains have remained rare, which is probably due to a considerable fitness cost that is associated with *vanA* in *S. aureus* [23].

Fluoroquinolones represent some of the most prescribed antibiotics. Fluoroquinolones exert their bactericidal effects by inhibiting the activity of topoisomerase II (gyrase) and topoisomerase IV, which are responsible for DNA superspiralization and respiralization, respectively. By inhibiting these activities, the synthesis of DNA is impaired and the bacterial cells die. Fluoroquinolones are effective against most MSSA infections but few MRSA infections [24]. *S. aureus* resistance to fluoroquinolones occurs via mutations in the *gyrA*, *gyrB* (topoisomerase II), *parC* (*grlA*), and *are* (topoisomerase IV) genes, and these mutations result in the synthesis of proteins with reduced susceptibility or insensitivity to fluoroquinolones [25]. Resistance can also be achieved through the overproduction of the chromosome-encoded proteins that are responsible for the efflux of fluoroquinolones through major facilitator superfamily (MFS) transporters [26].

The continual arms race between antibiotic discovery and the development of antibiotic resistance is a cause for great concern. The issue is exasperated by the high costs of novel antibiotic therapies, the limited effectiveness of antibiotic treatments, and the adverse outcomes for patients [27]. Alternatives to antibiotics are therefore needed to treat the devastating infections that are caused by antibiotic-resistant strains of *S. aureus*. Some potential therapeutics include bacteriophages, probiotics, prebiotics, and antimicrobial peptides that are produced by other bacteria (i.e., bacteriocins) [28,29,30,31]. The aim of this review is to provide a comprehensive evaluation of bacteriocins and focus on the translational potential of bacteriocins as an alternative therapeutic for *S. aureus* infections.

## 2. Bacteriocins

### 2.1. Background

Bacteriocins are a group of proteins or peptides that are produced by bacteria to kill competitors. They are usually mostly active against phylogenetically closely related strains. Bacteriocins are a diverse group consisting of various sizes, characteristics, producer species, target species, and mechanisms of action. The production of bacteriocins is costly to the producer cell, which is why their production is tightly regulated [32,33,34]. Despite the high cost to the producer cell, several isolates of many bacterial and archaebacterial species produce bacteriocins. It has been estimated that bacteriocin-producing isolates can be found in 30–99% of bacterial species [35,36,37]; therefore, it is believed that bacteriocins enhance the competitiveness of producer cells.

Bacteriocins can be classified into several categories. Both Class I and Class II bacteriocins are heat-stable and have a molecular weight of less than 10 kDa. Class I bacteriocins are post-translationally modified peptides that often result in non-standard amino acids, whereas Class II bacteriocins are not post-translationally modified. Class III bacteriocins, in contrast, are thermo-labile large (>30 kDa) unmodified proteins [37,38,39]. Due to the differences in size and heat lability between Class III bacteriocins and the other classes, some have suggested reclassifying Class III as bacteriolysins [37]. Class I and Class II bacteriocins can be further divided into three and four subclasses, respectively. Class Ia bacteriocins (i.e., the lantibiotics) characteristically contain lanthionine and β-methyl lanthionine, Class Ib comprise carbacylic lantibiotics that contain labyrinthin and labionin, and Class Ic bacteriocins are composed of sactibiotics that usually contain cysteine sulfur to α-carbon linkages [40,41,42]. With respect to the Class II subclasses, Class IIa contain pediocin-like bacteriocins, Class IIb contain two-component systems that form an active complex that is constructed from two different peptides, Class IIc contain circular bacteriocins, and Class IId contain non-pediocin-like bacteriocins [40,42].

The exact method of bacterial killing by bacteriocins varies, but it typically occurs in one of two ways: the bacteriocin interacts with the target cell envelope or it exerts its bactericidal activity inside the target cell, such as by inhibiting the synthesis of essential macromolecules. For example, nisin (nisin A) is a Class Ia bacteriocin, a lantibiotic, and one of the most extensively studied bacteriocins. It has a well-known mechanism of action against a wide spectrum of Gram-positive bacteria, including *S. aureus*. Nisin A, its biological and synthetic variants, and several other lantibiotics prevent cell wall synthesis and form pores in the target cell by binding to the peptidoglycan precursor lipid II molecules, which results in target cell death [37,43,44]. Interacting with lipid II is a common mechanism of bacterial killing among Class Ia bacteriocins [45,46,47,48], but the mechanism of action and the receptors involved in Class Ib and Class Ic bacteriocin-induced bacterial killing remain unknown [40]. The four Class II subclasses induce membrane permeabilization via pore formation with amphiphilic helical structures that can insert into the target cell membranes without the need for an “anchor” molecule like lipid II, which results in depolarization and target cell death [37]. However, the receptors involved in pore formation by Class II bacteriocins vary. In general, Class IIa bacteriocins utilize mannose permease, Class IIb use undecaprenyl pyrophosphate phosphatise (UppP), Class IIc involve an ABC transporter, and Class IId bacteriocins utilize a metallopeptidase [40]. Class III bacteriocins also result in target cell death by catalyzing the hydrolysis of the cell wall, but the specific receptors that are involved remain unknown [37]. Even though the mechanisms underlying the bactericidal activity of some bacteriocins have been identified, there are many more bacteriocins with mechanisms that have yet to be elucidated. 

### 2.2. The Strengths and Limitations of Bacteriocins

When employed as bactericidal agents, bacteriocins offer several advantages over antibiotics. Bacteriocins can be considered more “natural” to humans because many are produced by bacteria that are present in the foods that have been consumed throughout human history. For example, lactic acid bacteria (LAB) produce a wide range of bacteriocins that can be isolated from food products such as meat and cheese [49]. Bacteriocins are also stable at high temperatures and in environments with extreme pH levels, and some bacteriocins retain bactericidal activity even after autoclavation [50,51]. Compared with antibiotics, many bacteriocins have lower toxicity profiles [52], and some reach their desired effect at lower concentrations (i.e., are more potent) [53]; however, a few studies have observed signs of toxicity in certain cell lines from animal (e.g., mice, rabbits, cows) and human sources [54,55].

Bacteriocins may be produced by chemical synthesis, using cell-free systems, or by fermentation of the naturally producing or engineered bacteria [56]. Historically, the production of bacteriocins involved batch fermentation by a producer strain. Production and purification are often challenging and may be optimized by heterologous bacteriocin expression [57]. As for most antibiotics, the chemical synthesis of bacteriocins is challenging due to their often complex chemistry. Bioengineering has given rise to bacteriocin derivatives with enhanced antimicrobial activity and derivatives that are tailored for therapeutic interventions [28,58,59]. Despite advances in purification techniques and expression systems, the production of bacteriocins in higher quantities as needed for their potential use as therapeutics is still a challenging and expensive task [60,61,62].

Conventional antibiotic therapies tend to have a broad-spectrum nature; thus, commensal bacteria of the gut are often eliminated as unintended consequences of the treatment. The concomitant elimination of non-pathogenic members of the gut microbiome can lead to the overgrowth of opportunistic pathogens. For example, antibiotic exposure is one of the most important risk factors for developing *Clostridioides difficile* infection (CDI) [63,64]. Bacteriocins have somewhat narrower target spectra than antibiotics, but the mechanisms of action of bacteriocins are not sufficiently species-specific that the killing of non-pathogenic bacteria with potentially important roles for human health can be completely avoided. Furthermore, the narrower target spectrum of bacteriocins can become problematic in the clinic because it requires that clinicians identify the causative agent of infection. Initial antimicrobial therapy of critically ill patients typically includes broad-spectrum antibiotics due to the etiological agent often being unknown at the time, and the early administration of these treatments is an important determinant of hospital mortality [65,66,67,68]. The potential immunogenicity of these bacteriocin therapies is also of concern, especially for the longer peptides. 

Bacteriocins can be used to eliminate antibiotic-resistant strains because their mechanism of action differs from antibiotics [28]. However, it is to be anticipated that resistance to bacteriocins will develop or similarly lead to the resistance to antibiotics. Mechanisms of bacteriocin resistance are diverse and include enzymatic inactivation such as via proteolysis, target modifications, and efflux systems, among others. These resistance mechanisms have been elucidated mostly for nisin [69]. For example, acid-adapted *Listeria monocytogenes* displayed resistance to the bacteriocins nisin and lacticin 3147 [70], while *Streptococcus bovis* developed nisin resistance [71]. The spread of bacteriocin resistance is even more likely because bacteriocin-producing bacteria commonly have specific producer immunity factors encoded in their biosynthetic cluster; thus, specific resistance factors are already present in nature. The fact that bacteriocins commonly target bacteria that are phylogenetically related also means that such resistance factors may be readily functional once transferred by horizontal gene transfer to target pathogens. Moreover, target bacteria may develop non-specific resistance via other physiological alterations that are often based on random mutations, including the direct degradation of bacteriocins via enzymatic action, mimicry of natural immunity of the bacteriocin producer, and adaption of target cell membrane and/or growth conditions [72,73,74,75]. Finally, cross-resistance, or resistance to both antibiotics and bacteriocins, becomes a genuine and concerning possibility if bacteriocins and antibiotics are employed in combination as a dual therapeutic [76]. Unfortunately, there are a limited number of studies that have investigated the development of bacteriocin resistance in *S. aureus* [77], but Blake et al. reported a substantial decrease (4- to 32-fold reduction) in nisin susceptibility in small-colony variants of *S. aureus.* Comparative genome sequencing of a single nisin-resistant colony with a 32-fold increase in MIC values revealed two mutations, and the reintroduction of these mutations in nisin-susceptible strains conferred up to a 16-fold reduction in nisin susceptibility [78]. Thus, bacteriocins may represent a potential alternative to antibiotics for the treatment of *S. aureus* infections; however, the possibility of *S. aureus* developing or acquiring resistance cannot be understated.

Being peptides or peptide-like molecules, bacteriocins are generally susceptible to degradation by proteolytic enzymes (proteases), which usually results in a significant loss of antimicrobial activity. The sensitivity to digestive proteases (e.g., trypsin, chymotrypsin, and pepsin) is problematic when considering bacteriocins as potentially orally administered therapeutics [51,56,79,80]. Parenteral administration might avoid degradation of the bacteriocin in the digestive tract, but it cannot circumvent all bacteriocin–protease interactions as the bacteriocin will still be subjected to proteases in the blood [77,81]. Bacteriocins are rarely resistant to proteolytic enzyme degradation, but one study has shown that the bacteriocin BAC1B17 was resistant to multiple proteases, including digestive proteases, for up to two hours with no loss of antimicrobial activity against several MRSA strains [82]. Furthermore, many lantibiotics have increased anti-proteolytic stability due to their characteristic lanthionine bridges, which can be interpreted as an evolutionary adaptation to counteract the enzymatic inactivation by target organisms [83]. Enzymatic degradation dramatically decreases the bioavailability and half-life of bacteriocins, limiting the potential of bacteriocins to serve as a therapeutic alternative to antibiotics. 

To combat some of the innate limitations of bacteriocins, mainly the enzymatic degradation issue, delivery systems that use nanoparticles have been developed. Nanoparticles are composed of metals, metal oxides, inorganic matter, organic matter, and/or carbon that are <100 nm in size and have varying dimensionality [84,85,86]. Several in vitro studies involving nisin or other bacteriocins have shown that using nanoparticles to encapsulate bacteriocins can enhance their therapeutic effects (e.g., increasing antimicrobial killing, bioavailability, and half-life) and minimize their degradation. For example, by using a combination of chitosan and alginate as delivery vehicle, nisin-loaded nanoparticles had greater antimicrobial effects against *S. aureus* than free nisin and better inhibited the growth of *S. aureus* in milk and cheese [87,88]. Another study found that a nisin-loaded poly(vinyl alcohol)–wheat gluten–zirconia membrane had well-controlled nisin release and that the delivery method enhanced nisin’s antimicrobial activity against *S. aureus* [89]. In a murine excisional skin infection model, a nisin-containing poly(ethylene oxide) and poly(D, L-lactide) nanofiber (50:50) blend significantly reduced *S. aureus* burden after seven days compared with the no-nisin control nanofiber dressings [90]. Finally, Liu et al. used nanoparticles to deliver the hydrophobic bacteriocin micrococcin P1 that is produced by a strain of *Staphylococcus hominis* and achieved a significant reduction in the *S. aureus* CFU and disease manifestations in systemic mouse and skin infection models [91]. Thus, using nanotechnology to enhance the antimicrobial effects of bacteriocins and mitigate their limitations shows promise, but additional studies are needed to determine toxicity profiles, pharmacokinetics, and potential interactions between the nanoparticles, bacteriocins, and target bacteria.

## 3. In Vivo Experiments Evaluating the Efficacy of Bacteriocins against *S. aureus* Infections

### 3.1. Skin Infection Models

Skin and soft tissue infections (SSTIs) are common bacterial infections that account for ~10% of infection-related hospital admissions in the United States [92,93]. SSTIs vary in severity, ranging from mild superficial infections (e.g., folliculitis, cellulitis) to life-threatening infections such as necrotizing fasciitis and shock due to Staphylococcal scalded skin syndrome. Treatment depends on the infection severity, etiologic agent, and drug susceptibility of the pathogen [94,95,96]. *S. aureus* is a leading cause of SSTIs, and many of these infections involve drug-resistant strains of *S. aureus*. 

There are several in vivo studies that have investigated the translational potential of bacteriocins to treat *S. aureus*-induced SSTIs. Van Staden Adu et al. used a mouse wound infection model with imaging of the bioluminescent *S. aureus* Xen36 to test the antimicrobial activity of the bacteriocins nisin, clausin, and amyloliquecidin compared with mupirocin ointment. The latter is often used as a treatment for *S. aureus*-induced SSTIs in the clinic. The bioluminescence of *S. aureus* was significantly reduced in all three bacteriocin treatment groups to a value that was similar to that observed in the mupirocin treatment group [97]. In addition, the bacteriocin-treated mice had smaller wounds on day seven of the experiment with no significant difference in the numbers of *S. aureus* cells between the bacteriocin and antibiotic treatment groups [97]. Another study involving nisin found that *S. aureus* burdens were significantly reduced in mice that were treated with a nisin-eluting nanofiber wound dressing compared with the no-nisin control nanofiber dressing [90]. As antimicrobials often become ineffective in single-antimicrobial formations, Ovchinnikov et al. investigated the efficacy of a combination formulation of two bacteriocins (garvicin KS and micrococcin P1) and penicillin G to treat MRSA skin infections in a mouse model compared with fucidin cream, which is commonly used to treat skin infections. The three-component formulation eradicated *S. aureus* Xen31, a multidrug-resistant bioluminescent strain of MRSA, from skin puncture wounds after four consecutive days of treatment [98]. These results suggest that the addition of bacteriocins to previously ineffective antibiotics such as penicillin G might render these antibiotics effective again. However, any conclusions drawn from the study are limited because the in vivo efficacy of the individually administered penicillin G, garvicin KS, and micrococcin P1 and a bacteriocin-only combination of garvicin KS and micrococcin P1 were not tested. Finally, the recently discovered bacteriocin lugdunin, which is produced by strains of *Staphylococcus lugdunensis*, showed strong in vitro activity against *S. aureus* and some other tested Gram-positive pathogens [99]. In a skin infection model, application of lugdunin led to strongly reduced surface-attached *S. aureus* CFU but only moderately reduced the CFU in deeper skin tissue. These results indicate that an analysis of deeper tissue should generally be included in similar efficacy assessments of topical anti-*S. aureus* formulae.

Rather than using pure bacteriocin substance, bacteriocin-producing strains can also be topically applied to wound infections using a “probiotic” approach. Liu et al. applied a micrococcin P1-producing *S. hominis* strain to *S. aureus*-infected wounds in mice while using the bioluminescent strain Xen 36. The authors reported reduced *S. aureus* burden and wound closure time when using this approach [91].

### 3.2. Respiratory Infection Models

One of the many manifestations of *S. aureus* infection is pneumonia, which is usually seen in patients who are recovering from influenza. Individuals who are colonized on the skin or in the nares are the most at-risk for developing staphylococcal pneumonia [100]. Current treatment includes vancomycin or linezolid for MRSA and oxacillin, nafcillin, or cefazolin for MSSA [101,102]. As resistance to these antibiotics, particularly with respect to linezolid and vancomycin, continues to become more common in healthcare settings, alternative methods of abating *S. aureus* respiratory infections are needed; two bacteriocins, nisin F and NVB333, have shown some promise in this regard [103,104]. 

Nisin F, a natural derivative of nisin, is a lantibiotic produced by *Lactococcus lactis* F10 that has anti-staphylococcal activity in vitro [105]. One study evaluated if nisin F could treat immunosuppressed mice with *S. aureus*-induced respiratory infections. De Kwaadsteniet et al. found that nisin F prevented lung tissue damage caused by *S. aureus* in immunosuppressed mice, but nisin F itself had no effect on the healthy, non-immunosuppressed mice, as both treated and untreated mice had no lung tissue damage [104]. Boakes et al. evaluated NVB333 as a potential therapeutic in a murine lung infection model. They challenged mice with *S. aureus* MRSA UNT084-3 intranasally and gave NVB333 2 and 14 h post infection. NVB333 significantly reduced bacterial loads 26 h after infection compared with vancomycin treatment in a dose-dependent fashion [103]. However, NVB333 could not fully eradicate *S. aureus* from the lungs.

### 3.3. Systemic Infection and Other Severe Infection Models

Severe systemic *S. aureus* infection is often modeled using intraperitoneal injections. After an intraperitoneal challenge, large volumes of bacterial suspensions are quickly absorbed into the body. *S. aureus* subsequently invades the organs and causes systemic infection. Such models are often used to evaluate the effectiveness of novel therapeutics in preventing severe and rapid-onset infections. Several studies have used intraperitoneal injections, for example, to examine the effectiveness of mutacin 1140, lysostaphin, and lacticin NK34 against systemic MRSA infections. 

Mutacin 1140 is a member of the epidermin-type lantibiotics with a broad activity spectrum against Gram-positive bacteria. Like other bacteriocins, mutacin 1140’s clinical applications are limited by its short half-life (e.g., 1.6 h in a rat model) [106]. As it is speculated that the high rate of clearance is associated with enzymatic degradation and attack by nucleophiles [107], Geng et al. evaluated the effects of charged and dehydrated residues on mutacin 1140’s pharmacokinetics after an MRSA intraperitoneal challenge. The alanine substitutions for lysine (K2A) or arginine (R13A) resulted in significantly lower clearances and higher plasma concentrations of mutacin 1140 over time. A single intravenous injection (10 mg/kg) of the K2A and R13A analogs protected 100% of the mice, while a smaller dose (2.5 mg/kg) resulted in a 50% survival compared with a 100% mortality in the vehicle control group. In addition, there was a significant reduction in the bacterial load in the kidneys and livers of the 10 mg/kg analog groups compared with the vehicle control group [107]. While these results are promising, additional studies are needed to determine the most effective dosage, tolerability, toxicology profiles, and efficacy of infection clearance of mutacin 1140 versus standard-of-care antibiotic regiments. Two studies evaluated a Class III bacteriocin, lysostaphin, for the treatment of systemic *S. aureus* infections compared with the antibiotic vancomycin. Lysostaphin, which degrades the staphylococcal cell wall, was originally isolated from *Staphylococcus simulans* [108]. Treatment with lysostaphin was shown to be as effective as treatment with vancomycin in MRSA and MSSA rodent neonate infection models [109,110]. While these studies show that lysostaphin is a potent bacteriocin that can clear *S. aureus* infections, additional studies are needed to determine the extent of lysostaphin’s translational potential. Another study evaluated the efficacy of lacticin NK34 at treating systemic *S. aureus* infections. Lacticin NK34 is a nisin-like lantibiotic that was isolated from *Lactococcus lactis* and that has antimicrobial activity against *S. aureus* and other Gram-positive bacteria [111]. After an intraperitoneal challenge of *S. aureus* at the minimal lethal dose (MLD), Kim et al. treated mice with lacticin NK34. Lacticin NK34 treatment increased the survival between treatment groups from 5% in the non-treatment group to 72% in the treatment group after seven days [112]. However, the study only used a single strain of *S. aureus*, and a previous study published by the same laboratory showed that multiple *S. aureus* strains were not inhibited by lacticin NK34 [111].

Osteomyelitis, an infection of the bones, is commonly caused by staphylococcal species such as *S. aureus* and *S. epidermidis* [113]. To prevent orthopedic-related bacterial infections after surgical procedures, polymethylmethacrylate (PMMA) bone cement that is loaded with antibiotics is often used, as it has been shown to reduce the rate of infection in revision hip arthroplasty and primary hip arthroplasty by 40% and 50%, respectively [114,115]. One study evaluated if nisin F could control orthopedic-related bacterial infections when loaded onto cement that was subcutaneously implanted on the back of mice before infection with bioluminescent *S. aureus* Xen 36. The nisin F-loaded cement prevented *S. aureus* infection for seven days, and no viable bacterial cells were obtained after the mice were euthanized and the bone cylinders were removed [116]. 

*S. aureus* is also one of the most common causes of infective endocarditis, which has a high rate of mortality [117]. In one study, the lantibiotic NAI-107 (“microbisporicin” produced by *Microbispora* sp.) [118] was investigated by Jabés et al. as a treatment for both *S. aureus*-induced endocarditis and systemic infection. The authors used a granuloma pouch model with an intraperitoneal injection of a MRSA strain and NAI-107 that was intravenously administered. NAI-107’s bactericidal activity was dose-proportional, and a single 40 mg/kg dose caused a 3-log reduction in viable MRSA in exudates compared with two 20 mg/kg doses at 12 or 24 h, respectively [119]. In the same study, female rats were induced with endocarditis by *S. aureus* before treatment with NAI-107 or vancomycin (twice/day for five days). The authors found a dose-dependent reduction in the bacterial load, which was significantly lower than that obtained with vancomycin at higher doses. Overall, NAI-107 showed promise in its ability to severe *S. aureus* infections, although the study did not assess mortality [119].

### 3.4. Nasal, Intestinal, and Skin Carriage

*S. aureus* colonizes the anterior nares in about one third of the population [120]. Nasal colonization with *S. aureus* and especially with MRSA variants is a known risk factor for developing infections, including complications after surgery, and several studies have indicated that nasal *S. aureus* represent the source of the infectious bacteria [120,121,122,123]. The antibiotic mupirocin is often used to decolonize the nares from *S. aureus* prior to surgery and is occasionally combined with chlorhexidine body washes; however, given the rising abundance of mupirocin-resistant *S. aureus* strains, the development of alternative decolonization methods needs to be considered [124,125,126].

Epidermicin NI01 is a novel class II bacteriocin produced by *S. epidermidis* that exhibits antimicrobial activity against a wide range of pathogens, including MRSA [127]. Epidermicin has been the subject of extensive study [127] even though its exact mechanism of action remains unknown. One study showed that epidermicin NI01 was at least as effective as mupirocin at decolonizing MRSA from the nares of cotton mice [128]. Twice-daily administrations of mupirocin multiple days in a row were required to reduce colonization, while a single dose of epidermicin NI01 resulted in persistent decolonization that lasted for days. Additionally, Sandiford and Upton found that epidermicin NI01 was active against susceptible bacteria at lower concentrations and that exposure to the bacteriocin did not result in the development of resistance in the target bacteria [127]. These results suggest that epidermicin NI01 is a promising antibiotic alternative that can reduce the risk of developing MRSA infections in hospitals, but there are additional alternatives. For example, mersacidin is a lantibiotic that is produced by *Bacillus* sp. HIL Y-85, 54728. It was shown to inhibit the growth of MRSA and other Gram-positive bacteria in vitro [129]. Kruszewska et al. found that mersacidin eliminated *S. aureus* nasal colonization after twice-daily applications for three days in a mouse rhinitis model. No signs of systemic infection were present and no *S. aureus* cells were recovered from the nares of mice treated with mersacidin [130]. Finally, *S. lugdunensis* that produced the above-mentioned lugdunin but was not a non-producing isogenic *S. lugdunensis* outcompeted *S. aureus* when applied to the noses of cotton rats [99], suggesting the potential of a probiotic approach that uses such bacteriocin-producing bacteria to control *S. aureus* nasal carriage.

It should be noted that recent findings have emphasized the role of the intestinal carriage of *S. aureus*, as carriers of *S. aureus* have many more *S. aureus* in their gut than in their noses [3,131]. Elimination of intestinal carriage by *Bacillus subtilis*, which produces molecules that inhibit the quorum-sensing system that is essential for *S. aureus* intestinal colonization [30], have been suggested to control *S. aureus* intestinal colonization and, due to the key role attributed to the intestine as the source of *S. aureus*, *S. aureus* colonization in general [131]. On the other hand, bacteriocin-producing bacteria have been shown to eradicate intestinal pathogens in mouse colonization models where most of the natural microbiome was eradicated or in germ-free mice such as *Salmonella* sp. by microcin-producing *Escherichia coli* [132]. However, whether bacteriocins show sufficient target specificity for a “microbiome editing” probiotic approach in healthy people that leaves the microbiome virtually intact—as does the more targeted *B. subtilis* quorum-quenching approach for *S. aureus*—is questionable due to the often relatively broad target spectrum of bacteriocins. It may work in specific cases [133], but generally, such approaches should be limited to the treatment of severe disease where microbiome effects are a calculated risk. Possibly, the effects on the nasal microbiome when using *S. aureus*-targeted bacteriocins for nasal carriage are less problematic than those in a potential gut-targeted bacteriocin-based approach; however, similarly to the traditional mupirocin-based strategies, they suffer from the general problem of recolonization from non-targeted *S. aureus* sites of colonization, particularly the intestine. 

As for the skin, *S. aureus* does not normally colonize the skin of healthy individuals in considerable numbers, but increased *S. aureus* colonization is found in affected areas of atopic dermatitis (AD) patients [134]. Therefore, bacteriocin therapy has long been proposed as a means of *S. aureus*-targeted microbiome editing for AD treatment [135,136]; a recently completed clinical trial demonstrated the safety of a bacteriotherapy approach to treat AD with a bacteriocin-producing *Staphylococcus hominis* [137]. Furthermore, AD microbiome editing with an even more *S. aureus*-targeted form is possible with bacteriocins through quorum-quenching approaches in a form similar to that which was outlined above for the gut [138]. Of note, as the staphylococcal quorum-sensing signals are peptides, some have classified them as antimicrobial peptides [136]. However, they are not directly antimicrobial and thus clearly distinguished from bacteriocins.

A summary of in vivo experiments evaluating the efficacy of bacteriocins in *S. aureus* infections is shown in Table 1.

## 4. Additional Applications of Bacteriocins against *S. aureus*

### 4.1. Industrial Applications

Bacteriocins are used as antimicrobial agents in several sectors of industry. The use of bacteriocins in the preservation of meat, dairy, egg, and vegetable products has been thoroughly investigated, and bacteriocins are assumed to be nontoxic to the consumer [143,144,145]. The most widely studied bacteriocin, nisin (trademarked as Nisaplin^TM^), is used as a natural food protectant when preparing dairy products and canned foods in over 48 countries as it is active against a range of Gram-positive pathogens such as *S. aureus* and food-borne pathogens such as *Listeria monocytogenes* and *Clostridium botulinum* [146,147,148]. Other bacteriocins with anti-*S. aureus* activity that could be used in food preservation include enterocin AS-48 in skim milk, non-fat soft cheese, and vegetable sausages [149,150,151]; enterocin CCM 4231 in skim milk and yogurt [152]; and bacteriocin CAMT2 in meat products and milk [153,154].

The potential use of bacteriocins in industry is, however, most important in the agricultural sector. Antibiotics used in farm animals account for approximately 73% of all antibiotic use [147,155]; in the United States, that number rises to 80% [156]. Antibiotics are used in agriculture and aquaculture to treat infections in sick livestock animals, as prophylactics to prevent disease in healthy animals, and as growth promoters to improve weight gain and feed conversion [157,158,159]. Approximately 70% of the antibiotics used in farm animals and in human patients are identical or nearly identical [160], making agriculture a significant driving force in the development of drug-resistant pathogens [161]. It is therefore important to identify areas of the agricultural sector that could employ alternative therapies, such as bacteriocins, to minimize antibiotic use. 

For example, bacteriocins could be used to treat bovine mastitis, which is predominantly caused by *S. aureus* [162], in lactating and dry cows. Cao et al. investigated the ability of a nisin-based formulation vs. the antibiotic gentamicin, to which many *S. aureus* isolates have resistance, to treat bovine clinical mastitis in lactating cows. The nisin formulation offered a similar clinical cure rate to gentamicin [139]. Another study found that a bismuth-based teat seal product containing lacticin 3147 provided protection against *S. aureus*-induced bovine mastitis [142]. Several other bacteriocins have potential as treatments for *S. aureus*-induced bovine mastitis based on in vitro data involving *S. aureus* isolates from clinically presenting dairy cows [163,164]. Bacteriocins could also be used as prophylactics in lieu of antibiotics in the agricultural sector. Ryan et al. incorporated lacticin 3147 into an intramammary teat seal product for dry cows and found robust antimicrobial activity against all strains of streptococci and staphylococci that were tested, which included seventeen strains of *S. aureus*. The study also investigated a nisin-based intramammary teat seal product, but nisin’s insolubility at the physiological pH makes the nisin-based intramammary teat seal product unsuitable as a prophylactic for preventing bovine mastitis [165]. 

Bacteriocins have further potential applications in the pharmaceutical industry. For example, fermenticin HV6B is produced by a strain of *Lactobacillus fermentum* that was originally isolated from the human vagina. In vitro experiments showed that fermenticin HV6B possesses spermicidal activity, results in sperm immobilization, and is antimicrobial against several opportunistic pathogens, including *S. aureus* [166], but it has been noted that fermenticin HV6B is not suitable for commercial use because of its toxic effects on the native vaginal microbiota [167]. 

### 4.2. Bacteriocins against S. aureus in Clinical Trials

Results from pre-clinical animal studies and the successful use of bacteriocins in agriculture suggest that bacteriocins represent a viable alternative to antibiotics for treating *S. aureus* infections in humans. Unfortunately, to date, there are only a limited number of completed or ongoing clinical trials investigating the efficacy and safety of bacteriocins (Table 2). There are even fewer studies evaluating bacteriocins as a potential therapeutic for *S. aureus* infections (Table 2), and many studies have yet to publish or make their results available. Table 3 lists all bacteriocins, their structure and modes of action, if known, that were discussed in this review.

Novacta Biosystems Limited (UK) initiated a phase I clinical trial that investigated the safety, tolerability, and pharmacokinetics of single and multiple ascending orally administered doses of NVB302, which has promising in vitro and in vivo activity against MRSA and VRSA strains [103]. In part A of the study, up to five cohorts of eight healthy participants were randomized to receive either a single dose of NVB203 (starting dose: 100 mg) or a placebo. In part B of the study, up to four cohorts of eight healthy participants received a daily dose of NVB203 or a placebo for ten days. The trial was completed in 2012, but the results are not yet available (EudraCT/CTIS: 2011-002703-14). 

Most clinical trials investigating the ability of bacteriocins as an alternative therapy against *S. aureus* involve nisin. As *S. aureus* is the predominant etiological agent of acute mastitis in humans [168,169], Fernández et al. investigated the usage of a nisin solution in lactating women with clinical signs of staphylococcal mastitis. The participants were randomly assigned to the nisin solution (6 μg/mL) group or control group (a similar solution devoid of nisin), and the solutions were applied to the nipple and mammary areola for two weeks. Compared with day 0, where the staphylococcal counts in the breast milk between the two groups were similar, the counts were significantly lower on day 14 in the nisin group (3.22 ± 0.43 log_10_ CFU/mL) compared with the control group (5.01 ± 0.21 log_10_ CFU/mL). In addition, no clinical signs of mastitis were reported for the nisin group on day 14, while the control group had clinical signs for the entire study. Another clinical trial (NCT02928042) investigated the inhibitory effects of nisin and LAB members on the etiological agents of ventilator-associated pneumonia [170]. Several pathogens (i.e., *P. aeruginosa*, *A. baumannii*, *K. pneumonia*, and *S. aureus*) were obtained from the tracheal aspiration cultures of 80 patients who were treated with mechanical ventilation (>48 h). The probiotic effects of the LAB species on several *P. aeruginosa* strains were published in 2014 [171], but the results pertaining to nisin’s antimicrobial activity against *S. aureus* have not been made available. A different clinical trial (NCT02467972) evaluated how the addition of several experimental additives that included nisin could impact hunger, satiety, bowel movements, and colonic bacterial populations. Sixty-one healthy participants were recruited and randomly assigned to one of the experimental groups or the control group. Nisin was added to 18 portions of ready-to-eat frozen soups (3 meals/week over 6 weeks), and changes in colonic bacterial populations were measured using the microbiome with r16S DNA sequencing; however, the results have yet to be published [172]. antibiotics-12-01256-t002_Table 2Table 2Clinical trials with bacteriocins against *S. aureus*.Identifier or CitationInterventionProposed Sample Size/EnrollmentPrimary Endpoint(s)StatusResultsFernández, et al. [173]Application of a nisin solution (6 μg/mL) to the nipple and mammary areola8 lactating women with clinical signs of staphylococcal mastitis Evaluate the clinical signs of mastitis and bacterial loads after 2 weeksCompleteBacterial load in the nisin group was statistically lower than the control group. No clinical signs of staphylococcal mastitis were observed in the nisin group on day 14 of the trialClinical Trials.govNCT02467972One of the experimental groups had a dietary supplement of nisin. Nisin was added to ready-to-eat frozen soups (3 times/week with 18 portions in total)61 healthy participants Change in bowel function, colonic bacteria population, and hormonal parameters related to hunger and satiety after six weeksCompleteNot availableClinical Trials.govNCT02928042Tracheal aspiration-derived pathogens were treated with LAB members and nisin to evaluate their effectiveness at treating ventilator-associated pneumonia80 patients who were mechanically ventilated for at least 48 h were recruited and who had their tracheal aspirate cultures used for the studyThe antimicrobial properties and effects of nisin and LAB members on *P. aeruginosa*, *A. baumannii*, *K. pneumonia*, and *S. aureus* growth rateCompleteNot availableEudraCT/CTIS2011-002703-14Part A: a single dose (100 mg) of NVB302 or a placebo Part B: Once daily doses of NVB302 or a placebo for ten daysPart A: Up to five cohorts of 8 healthy subjectsPart B: Up to four cohorts of 8 healthy subjectsAssessment of the safety and tolerability of single and multiple oral ascending doses of NVB302CompleteNot available

## 5. Conclusions

Antibiotic resistance represents an urgent and growing threat to animal and human health worldwide. The arms race between antibiotic discovery and the emergence of resistant organisms is ever-ongoing, but investments into antibiotic development by biotechnology and pharmaceutical industries are slowing down [174]. *S. aureus* is a major contributor to antibiotic-resistant infections. Infections caused by antibiotic-resistant strains of *S. aureus,* such as MRSA, result in significant health-related and economic burdens across the clinical and agricultural sectors; thus, there is an urgent need to identify alternative pathways to treat these devastating infections. Bacteriocins represent one such alternative therapy. Bacteriocins are generally considered to be nontoxic and to have a narrower target spectrum than antibiotics, which reduces off-target effects on host mammalian cells and commensal bacteria. Nanotechnology can be used to enhance a bacteriocin’s antimicrobial activity against its target bacteria and minimize its limitations. Despite these benefits, there are few in vivo studies investigating the ability of bacteriocins as a potential therapeutic to treat *S. aureus* infections. 

As of 2021, Benítez-Chao et al. estimate that approximately 50% of studies investigating the antimicrobial action of bacteriocins in murine models lack toxicity and/or biosafety studies [175]. However, toxicity assays and biosafety studies are important precursors for initiating a clinical trial. For the in vitro and in vivo studies to translate into the clinic and for the therapeutic potential of bacteriocins to be maximized, future experiments must include these crucial data. In addition, there is a clear lack of clinical trials that assess systemically administered bacteriocins for the treatment of severe *S. aureus* infections, which can probably be attributed to the limitations that are associated with systemic administration, such as proteolytic degradation, toxicity, or immunogenicity. These problems may have prevented seemingly promising bacteriocins from proceeding to clinical trials, and the negative outcomes of pre-clinical assessments may not have been published. Even though the efficacy results from many published animal models of infection are promising, additional in vivo studies and clinical trials are needed before bacteriocins can become a truly viable alternative to antibiotics in the treatment of infections with *S. aureus*.
antibiotics-12-01256-t003_Table 3Table 3Additional information about bacteriocins of interest.BacteriocinPeptide SequenceMechanism of ActionReferenceMutacin 1140
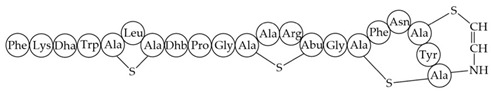
Inhibition of cell wall synthesis, pore formation[107,176,177,178]Epidermicin NI01MAAFMKLIQFLATLGQKYVSLAWKHKGTILKWINAGQSFEWIYKQIKKLWAR (unmodified)Multi-mode pore formation induced by four-helices[127,179]Nisin (Nisin A)
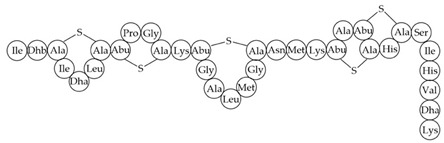
Nisin F: Nisin A H27N; Nisin Z: Nisin A H27N, I30VInhibition of cell wall synthesis, pore formation[105,180,181,182,183]NAI-107 (Microbisporicin)
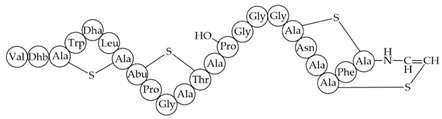
Inhibition of cell wall synthesis[118]CMB001
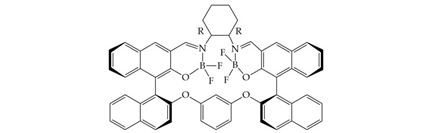
Unknown[141,184]Garvicin KSThree-component:GakA, MGAIIKAGAKIVGKGVLGGGASWLGWNVGEKIWKGakB, MGAIIKAGAKIIGKGLLGGAAGGATYGGLKKIFGGakC, MGAIIKAGAKIVGKGALTGGGVWLAEKLFGGK(unmodified)Unknown[185]Micrococcin P1
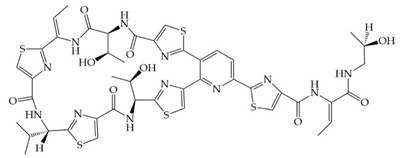
Inhibition of ribosomal protein synthesis[186]Lysostaphin26.9 kDa protein in mature formCleaving of pentaglycine bridges in the cell wall[187,188]NVB333
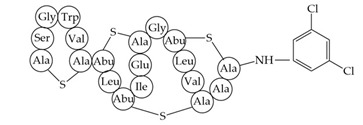
Inhibition of cell wall synthesis[103]Clausin
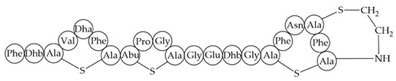
Inhibition of cell wall synthesis[47]Amyloliquecidin
Two-component lantibiotic, exact structure unknown, primary amino acid sequences:CAWYDISCKLGNKGAWCTLTVECQSSCN, TTPSSLPCGVFVTAAFCPSTKCTSSC
Inhibition of cell wall synthesis[48]Mersacidin
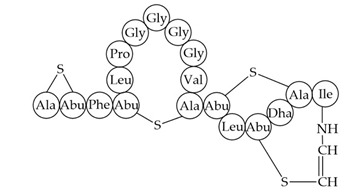
Inhibition of cell wall synthesis[45,189]Lacticin NK34
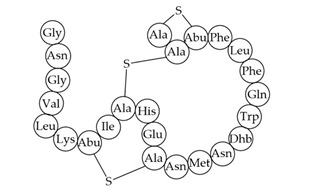
Inhibition of cell wall synthesis[112]Lacticin 3147
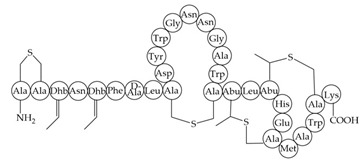
Inhibition of cell wall synthesis[46,190]Lugdunin
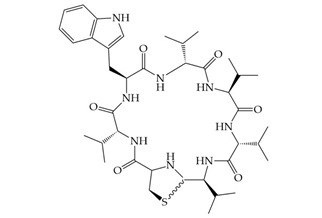
Disrupts membrane potential[99]

## Figures and Tables

**Figure 1 antibiotics-12-01256-f001:**
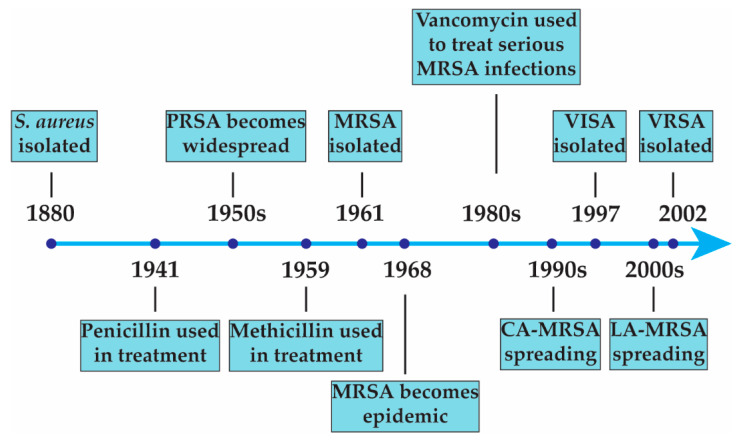
Emergence of antibiotic-resistant *S. aureus* strains.

**Table 1 antibiotics-12-01256-t001:** In vivo experiments evaluating the efficacy of bacteriocins in *S. aureus* infections.

Bacteriocin	Producing Strain	Target Strains	Model Organism	In Vivo Demonstration	Reference
Mutacin 1140	*Streptococcus mutans*	*S. aureus* (ATCC 25923 and ATCC 33591)	BALB/c mouse	Systemic infection	[107]
Epidermicin NI01	*S. epidermidis*	MRSA ATCC 43300	SPF Cotton rat	Nasal carriage	[128]
Nisin (Nisin A)	*Lactococcus lactis*	*S. aureus* Xen36	Nude mouse	Skin infection	[97]
			BALB/c mouse		[90]
Nisin F	*Lactococcus lactis* F10	*S. aureus* K *S. aureus* Xen36	Wistar rat BALB/c mouse	Respiratory infection Osteomyelitis	[104,116]
Nisin Z	*Lactococcus lactis*	Undetermined *S. aureus* strains	Cow	Mastitis	[139]
NAI-107	*Microbispora* ATCC PTA-5024	*S. aureus* 1524 *S. aureus* (ATCC 29213, USA200, 307109, MW2, USA300, ATCC 25923, 6538P, Smith, WIS-1)	SD rat ICR mouse	Endocarditis	[119]
				Intramuscular infection	[140]
CMB001	*Paenibacillus kyungheensis*	*S. aureus* USA300	ICR mouse	Intramuscular infection	[141]
Garvicin KS	*Lactococcus garvieae* KS1546	*S. aureus* Xen31	BALB/c mouse	Skin infection	[98]
Micrococcin P1	*Staphylococcus equorum* WS 2733	*S. aureus* Xen31	BALB/c mouse	Skin infection	[98]
Lysostaphin	*Staphylococcus simulans*	*S. aureus* USA300	Neonatal Wistar mouse Neonatal FVB mouse	Systemic infection Systemic infection	[109]
					[110]
NVB333	*Actinoplanes liguriae*	*S. aureus* UNT103-3, *S. aureus* ATCC 33591, MRSA UNT084-3	SPF CD-1 mouse	Intramuscular infection Respiratory infection	[103]
Clausin	*Alkalihalobacillus clausii*	*S. aureus* Xen36	Nude mouse	Skin Infection	[97]
Amyloliquecidin	*Bacillus velezensis*	*S. aureus* Xen36	Nude mouse	Skin infection	[97]
Mersacidin	*Bacillus* sp. *strain HIL Y-85,54728.*	*S. aureus* 99308	BALB/c mouse	Nasal carriage	[130]
Lacticin NK34	*Lactococcus lactis*	*S. aureus* 69	ICR mouse	Systemic infection	[112]
Lacticin 3147	*Lactococcus lactis* subsp. *lactis* DPC3147	Undetermined *S. aureus* strains	Cow	Mastitis	[142]
Lugdunin	*Staphylococcus lugunensis*	*S. aureus* USA3000	Cotton rat	Skin infection	[99]

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
