# Peer review of "Evaluating the Translational Potential of Bacteriocins as an Alternative Treatment for Staphylococcus aureus Infections in Animals and Humans"

_antibiotics, 2023, doi:10.3390/antibiotics12081256_

Round 1
Reviewer 1 Report
This is a well-written review that provides a comprehensive and up-to-date assessment of animal studies on bacteriocins against Staphylococcus aureus and their potential for translation into clinical practice. The authors provide a strong discussion on the strengths and limitations of bacteriocin treatments in comparison to conventional antibiotic therapies and identify gaps in the field. My only suggestion is for the authors to provide an updated discussion on the classification of bacteriocins and the current knowledge on the very diverse mechanisms of action of bacteriocins (e.g., receptors involved in activity apart from lipid II) since the authors are citing an outdated 2005 review for this section (Line 133-138).
Author Response
Thank you very much for your nice comments. We reworked the section of the manuscript that you indicated.
Reviewer 2 Report
The review by Heinzinger et al. covers extensively the current and foreseen applications of bacteriocins as alternatives to classical antibiotics in the perspective of increased spread of antibiotic resistance by S. aureus. The review is logically organized and easy to read. It contains useful information on clinical trials and suggested industrial applications. It also briefly discusses some of the sources of resistance that may appear. In this context I would have appreciated additional information on the structure of the bacteriocins mentioned in the text, together with some indications, when known, of their suggested mechanism of action, that will largely determine the risk of acquired resistance.
I would suggest that the authors add a table with the peptide sequences of the smaller bacteriocins (say those having less than 40-50 residues), their suggested mechanism (e.g. known receptor, cell wall synthesis, bilayer disruption…), resistance mechanisms if known (proteolysis, specific resistance factors in producing bacteria…). This table could be in the main text or as supplementary material.
Author Response
We added a table showing structure and mode of action of the bacteriocins mentioned in the review. We did not add a specific column about modes of resistance because these have been described virtually exclusively for nisin. Some additional information and reference to a review on nisin resistance was added to the text.